# Dietary Habits, Selenium, Copper, Zinc and Total Antioxidant Status in Serum in Relation to Cognitive Functions of Patients with Alzheimer’s Disease

**DOI:** 10.3390/nu13020287

**Published:** 2021-01-20

**Authors:** Katarzyna Socha, Katarzyna Klimiuk, Sylwia K. Naliwajko, Jolanta Soroczyńska, Anna Puścion-Jakubik, Renata Markiewicz-Żukowska, Jan Kochanowicz

**Affiliations:** 1Department of Bromatology, Faculty of Pharmacy with the Division of Laboratory Medicine, Medical University of Białystok, Mickiewicza 2D Street, 15-222 Białystok, Poland; sylwia.naliwajko@umb.edu.pl (S.K.N.); jolanta.soroczynska@umb.edu.pl (J.S.); anna.puscion-jakubik@umb.edu.pl (A.P.-J.); renmar@poczta.onet.pl (R.M.-Ż.); 2Podlasie Center of Psychogeriatrics, Swobodna 38 Street, 15-756 Białystok, Poland; litwk@wp.pl; 3Department of Neurology, Medical University of Białystok, M. Skłodowskiej-Curie 24a Street, 15-276 Białystok, Poland; kochanowicz@vp.pl

**Keywords:** Alzheimer’s disease, selenium, copper, zinc, total antioxidant status, diet

## Abstract

Oxidative stress plays a crucial role in the neurodegenerative process and can impair cognitive functions. In the prevention of Alzheimer’s disease (AD), an adequate consumption of dietary antioxidants may be a major factor. The objective of the study was to estimate selenium (Se), copper (Cu), zinc (Zn), and total antioxidant status (TAS) in the serum of patients with AD in relation to their cognitive functions and dietary habits. A total of 110 patients (aged 54–93 years) with early or moderate AD, as well as 60 healthy people (aged 52–83 years) were studied. The severity of the disease was assessed using the mini-mental state examination (MMSE) scale. Food-frequency questionnaires were implemented to collect the dietary data. The concentrations of Se, Cu and Zn in the sera were determined by the atomic absorption spectrometry method. TAS was estimated spectrophotometrically using ready-made kits (Randox). Significantly lower concentrations of Se, Zn and TAS, and higher Cu:Zn ratio in the serum of patients with AD, compared to healthy people, were observed. A low correlation between the MMSE score and TAS in the serum of AD patients and significantly higher MMSE values in patients with TAS above the reference range were also noted. In patients with serum Cu concentration above the norm, significantly lower MMSE values were found. Selected dietary habits such as the frequency of consumption of various food products had a significant impact on the concentration of the assessed parameters in the serum of people with AD.

## 1. Introduction

Alzheimer’s disease (AD) is a progressive, incurable, neurodegenerative disease, which is the most common form of dementia in the elderly. It most often occurs in people over 65, but can also affect younger people. Prior to the first symptoms, the disease may progress undetected for years. The average life expectancy after diagnosis is 7–10 years, and only fewer than 3% of patients survive longer than 14 years. It is forecasted that due to demographic trends (aging of the population), the incidence of AD will increase. Data from the Central Statistical Office (Warsaw, Poland) show that the population of people over 65 currently accounts for about 14.7% of the total population and in 2050, it will increase to 30%. In Poland, almost 400,000 people suffer from AD, and in 2050, more than one million people will have the condition [1], which will be a huge challenge for the healthcare system. The causes of the disease have not been understood so far. AD sometimes is hereditary; hence, some genetic predisposition is indicated [2]. Attempts have been made to explain its cause and several hypotheses have been proposed. The cholinergic hypothesis is the oldest. It takes into account reduced synthesis of the acetylcholine neurotransmitter. In 1991, the amyloid hypothesis appeared, suggesting that β-amyloid (βA) deposits play a fundamental role in the pathogenesis of the disease. This suggestion is supported by the position of the amyloid precursor protein gene (APP) on chromosome 21. In addition, the specific isoform of apolipoprotein, APOE4, is one of the major genetic risk factors for AD. Apolipoprotein increases the breakdown of βA, however some of its forms, such as APOE4, perform this task ineffectively. As a result, excessive amyloid accumulates in the brain. According to the hypothesis, updated in 2009, the amyloid-based mechanism that reduces neural connections in the brain in the rapid growth phase at the beginning of an individual’s life can be activated by aging-related processes, and leads to death of neurons, which is characteristic of AD. Another hypothesis assumes that it is tau protein abnormalities that contribute to the pathogenesis of the disease. In this model, the hyperphosphorylated tau protein strands begin to connect with each other, creating neurofibrillary tangles (NFTs) inside the neuronal bodies. This causes disintegration of microtubules, destroying the neuronal transport system, which leads to the impaired functioning of biochemical communication between neurons, and then death of the cells. Another hypothesis states that AD can be caused by age-related myelin breakdown in the brain [3,4,5,6]. Oxidative stress plays an important role in the neurodegenerative process and may precipitate cognitive decline. Studies have shown that the levels of various antioxidants are decreased in different age-related degenerative diseases, and thus oxidative stress may play a central role in the pathogenesis of many disorders that include neuronal degeneration, such as AD [7,8,9].

Selenium (Se) is an essential micronutrient in the diet due to the requirement for selenocysteine in some selenoproteins. This trace element is known to provide protection from reactive oxygen species (ROS)-induced cell damage, mainly thanks to the functions of glutathione peroxidases and selenoprotein P [10]. The brain is the last organ where Se deficiency appears, but also the first one to reach adequate levels of Se when the element is supplied. This demonstrates the importance of this antioxidant in brain functioning [11]. Glutathione peroxidase (GPx) is the first line of defense against free radicals, acting mainly against hydrogen peroxide and lipid peroxidation, protecting the brain against oxidative stress, which has an important role in AD. Se deficiency impairs GPx, resulting in oxidative stress. The most important selenoprotein for brain functions is selenoprotein P, synthesized at the cerebral level, which protects the brain against oxidative damage [12,13]. Therefore, Se may affect the progression of the disease, directly as an antioxidant or indirectly by improving metabolism in the brain [14].

Zinc (Zn) is necessary for the proper functioning of the immune system and protection of cells against oxidative stress [15]. This mineral component plays an important role in the brain due to its function as a neurotransmitter/neuromodulator [16]. Studies conducted among the elderly show a positive relationship between serum Zn concentration and selected cognitive functions. Moreover, a reduced risk of depression has been observed at higher Zn concentrations [17].

Copper (Cu) is of major significance in controlling the level of oxygen radicals as an active center of enzymes, such as oxidase and oxygenase, while Zn is directly involved in antioxidant activity and detoxification processes. Therefore, serum Cu and Zn can be considered markers of oxidative stress. Zn and Cu compete for the binding places of certain transporters and metal-binding proteins, which results in a negative relationship, where a low level of Zn means increased Cu concentration [18]. In AD, abnormal homeostasis of trace elements has been regarded as a contributing factor [19]. Although the results are inconclusive, Cu and Zn seem to show an antagonistic correlation with senile plaque load and decrease in the cerebrospinal fluid (CSF) of AD patients [20]. The accumulation of Cu and Fe in the brain has been connected with increased oxidative stress in AD [21]. The oxidative damage to NFTs, senile plaques, nucleic acids, and increased lipid peroxidation may be caused by a disruption in the homeostasis of redox-active metals: Cu and Fe [22]. Cu, abundant in senile plaques, is a mediator of the highly reactive hydroxyl radical (OH•) and thus contributes to an increase in oxidative stress in AD [23]. Metal homeostasis can be challenged in a number of ways: toxic elements such as aluminum (Al), lead (Pb), and mercury (Hg) can find their way into the human body through the environment, dietary habits, and smoking cigarettes, causing interactions.

Currently, there are no treatment methods that would completely cure AD. It is only possible to slow down the course of the disease and alleviate some symptoms. Due to the fact that pharmacological agents used in AD have numerous side effects, it seems advisable to seek other methods, including dietary interventions, which could improve patients’ quality of life.

The objective of this study was to estimate selenium, copper, zinc, and total antioxidant status in the serum of patients with AD in relation to their cognitive functions and dietary habits.

## 2. Materials and Methods

### 2.1. Characteristic of the Examined Groups

The studied group comprised 110 patients (aged 54–93 years) with early or moderate AD, diagnosed by a geriatrician, under the care of the Podlasie Center of Psychogeriatrics in Białystok and the Department of Neurology, Medical University of Białystok (Poland). The patients were diagnosed according to the 1984 criteria of the National Institute of Neurologic, Communicative Disorders and Stroke/Alzheimer’s Disease and Related Disorders Association (NINCDS-ADRDA), revised in 2007 [24]. The control group consisted of 60 healthy people, without cognitive impairment, most of whom were still professionally active, aged 52–83 years. Exclusion criteria from participation in the study included comorbidities, such as cancer, autoimmune diseases, and type 1 and type 2 diabetes. The severity of the disease was assessed using the mini-mental state examination (MMSE) scale. This method of assessing cognitive function is widely used among the elderly. It includes tests of orientation, attention, memory, language, and visual-spatial skills, on a 0–30-point scale. Characteristics of the study groups are presented in Table 1.

To collect the dietary data, food-frequency questionnaires (FFQ), developed by the Committee of Human Nutrition Science, Polish Academy of Sciences, were implemented. Patients with AD or their guardians, depending on the state of health, were asked to complete a questionnaire concerning the frequency of consumption of different food products. The questionnaires were completed with the participation of a geriatrician. The list consisted of 36 groups of food items (bacon, beer, butter, cereal products, coffee, cottage cheese, eggs, fresh fish, fresh vegetables, fruit, grain products, ham, honey, margarine, marmalade, meat products, meat, milk, offal, other kinds of cheese, potatoes, poultry, processed vegetables, pulses, sausages, soft drinks, sugar added to beverages, sweets, tea, tinned fish, tinned meat, vegetable oils, vodka, white bread, wholegrain bread, and wine). Frequent consumption was defined as eating certain food products from twelve to thirty times per month, except fish, in the case of which four to twelve times a month was regarded as frequent. Food products consumed less frequently were classified in the “sporadic consumption” group [25]. The protocol of the study was approved by the Local Ethical Committee (R-I-002/210/2018). All participants provided written consent to take part in the study.

### 2.2. Collection and Preparation of Samples

Blood samples (approximately 6 mL) from study participants were drawn using the vacutainer system test tubes containing a clot activator, recommended for elemental analysis (Becton Dickinson, France). The samples were centrifuged for 10 min at approximately 1000× *g*. Sera were acquired and stored frozen at −20 °C. The serum samples were deproteinated using 1 mol/L spectral-grade nitric acid (Merck, Germany) and 1% Triton X-100 was added, then mixed using vortex, and centrifuged for 10 min. The concentration of Zn was determined in the supernatant and the content of Cu was determined after dilution in 0.1 mol/L nitric acid. The concentration of Se was determined directly after dilution 1:1 with 0.2% Triton X-100.

### 2.3. Determination of Mineral Components and Total Antioxidant Status

The concentrations of Se, Cu and Zn in the serum were determined by the electrothermal (Se, Cu) and flame (Zn) atomic absorption spectrometry methods with Zeeman background correction at 196 nm, 324.8 nm, and 213.9 nm wavelengths, respectively (Z-2000 instrument, Hitachi, Tokyo, Japan). Working solutions for the calibration curves were made from standard solutions 1 g/L (Merck, Darmstadt, Germany). The limit of detection for Se was 1.86 µg/L, for Cu 0.53 µg/L and for Zn 0.013 mg/L.

Certified matrix reference material of human serum (Seronorm Trace Elements, Serum Level 1, 0903106, Sero AS, Billingstad, Norway) was used to test the accuracy and precision of the analytical techniques. All the results of control samples corresponded with the reference values. The precision of the methods for the determination of Se, Cu and Zn was 3.2%, 2.4% and 1.9%, respectively.

The Department of Bromatology, Medical University of Bialystok participates in a quality control program for trace element analysis supervised by the National Institute of Public Health–National Institute of Hygiene and the Institute of Nuclear Chemistry and Physics (Warsaw, Poland).

The measured concentrations of Se, Cu, Zn in the serum were interpreted in relation to the reference values: 66–104 µg/L, 0.7–1.6 mg/L and 0.7–1.3 mg/L, respectively [26].

The molar ratio of Cu:Zn was calculated using Microsoft Excel software.

Total antioxidant status (TAS) in the serum was measured spectrophotometrically at 600 nm wavelength, using the ready-made sets of tests by Randox Laboratories Ltd. (Crumlin, UK) and a UV–Vis spectrophotometer (Cintra 3030, GBC, Melbourne, Australia). ABTS^®^ (2,2′-Azino-di-[3-ethylbenzthiazoline sulphonate]) was incubated with peroxidase (metmyoglobin) and H_2_O_2_ to produce the radical cation ABTS^®^*+. It has a relatively stable blue-green color, which is measured at 600 nm. Antioxidants in the added sample caused suppression of this color production to a degree, which was proportional to their concentration. The accuracy of the method was verified using the TAS Control kit (Randox Laboratories Ltd., Crumlin, UK). The TAS reference range in the serum was from 1.3 to 1.77 mmol/L [27].

### 2.4. Statistical Analysis

Statistical analyses were performed using Statistica v.13.0 software (TIBCO Software Inc., Palo Alto, CA, USA). Data were tested for normal distribution by the Kolmogorov–Smirnov and the Shapiro–Wilk tests. Differences between independent groups were tested by the Mann–Whitney U test or the ANOVA Kruskal–Wallis test, depending on the number of groups. Correlations were calculated by the Spearman rank test. To estimate the influence of dietary habits on Se, Cu, and Zn levels and TAS in the examined patients, a stepwise multiple linear regression analysis was used. Differences at the significance level of *p* < 0.05 were considered statistically significant.

## 3. Results

In our study, 73% of the AD patients were women. We observed significantly lower concentrations of Se and Zn in the serum of patients with AD (69.1 ± 19.3 µg/L; 0.75 ± 0.33 mg/L) compared to healthy people (79.8 ± 22.0 µg/L; 0.92 ± 0.23 mg/L). Additionally, women with AD had significantly lower serum levels of these micronutrients compared to men. The concentration of Cu in the serum of AD patients did not differ significantly from the control group, but we observed differences depending on the gender. On the other hand, the Cu:Zn molar ratio was significantly higher in AD patients, and in women with AD, this ratio was significantly higher compared to men and women from the control group. We observed significantly lower TAS in the serum of patients with AD (1.11 ± 0.42 mmol/L) compared to healthy people (1.43 ± 0.67 mmol/L). Additionally, there were significant differences in TAS concentrations depending on gender. The results are presented in Table 2.

We did not note any significant differences in the concentration of the examined parameters depending on cigarette smoking status. Additionally, we did not observe any significant correlations between BMI or the age of patients and the concentrations of Se, Cu, Zn and TAS in the serum.

The average MMSE score of the examined patients was 20.4 ± 4.3, which indicates mild dementia. We observed a positive low, but significant correlation between the MMSE scores and TAS in the serum of patients with AD (r = 0.21, *p* < 0.05). Additionally, patients with early stages of the disease had significantly higher (*p* < 0.003) TAS compared to patients with moderate AD (1.33 ± 0.46 mmol/L and 1.05 ± 0.35 mmol/L, respectively). The data are not presented in the tables. There were no significant differences in the concentrations of the examined micronutrients and in the Cu:Zn values between the groups of patients with early and moderate AD. After dividing the patients with AD according to the concentration of the tested micronutrients and TAS in the serum into three groups—below, within the normal range and above the reference values—significant relationships were shown between the concentrations of Cu and TAS in the sera and the values of MMSE. The lowest MMSE values were found in patients with serum Cu concentrations above the norm, while in the group with serum TAS values above the reference range, significantly higher MMSE values were found (Figure 1 and Figure 2).

Stepwise multiple linear regression analysis showed that selected dietary habits, from 22% in the case of Se to 38% in the case of Cu, may positively or negatively affect the concentrations of these micronutrients in the serum. The analysis showed that the frequency of consumption of four food products could determine the concentration of TAS in the serum in 15% of participants. A detailed list of products included in the regression models is presented in Table 3. The results did not meet the criteria of normal distribution; therefore, they are expressed as the median, lower, and upper quartiles. For easier comparison with the results of other authors, they are presented as average values and standard deviations.

## 4. Discussion

Redox imbalance, which is a result of excessive ROS generation or an antioxidant system disruption, triggers oxidative stress [23]. Induction of oxidative stress and inflammation associated with age can lead to neurodegenerative processes and, consequently, cognitive impairment. Oxidative stress and neuroinflammation induced by βA protein deposits in the brain are considered to be important factors in the pathogenesis of AD. βA and tau are metal-binding proteins. Microelements, such as Se, Zn, and Cu, have a significant impact on brain function, although altered homeostasis and distribution can be associated with neurodegenerative diseases. Abnormal metabolism of trace elements can affect synaptic signaling pathways and mitochondrial functions, causing protein aggregation, oxidative stress, and inflammation, leading to synaptic dysfunction and a reduction in neurons in the brain of AD patients [28].

In the present study, we have observed significantly lower concentrations of Se and Zn in the serum of patients with AD than in healthy people, as well as lower serum concentrations of these micronutrients in women compared to men. However, all median concentration values were within the reference ranges [26]. The results are consistent with some studies of other authors. AD patients had significantly lower Se levels in plasma than healthy people [29,30]. However, in the work of Xu et al., no differences in the concentration of Se in the plasma of AD subjects compared to healthy control was shown. There were also no differences in Se concentrations depending on gender [31]. On the other hand, González-Domínguez et al. demonstrated significantly lower serum Se concentrations in AD patients compared to those in persons with mild cognitive impairments [32]. Research indicates that the relationship between Se and AD may mean two things: (i) Se can be reduced due to the oxidation that accompanies aging and progress of AD; and (ii) change in Se levels may be linked to dietary intake [33]. There is a shortage of research correlating Se status and AD progression, and most of the relevant papers have only estimated Se levels in plasma or whole blood. The available research into Se in AD patients is inconclusive. Ceballos-Picot et al. [34] observed higher Se plasma and erythrocyte levels in AD patients when compared with a control group. However, Smorgon et al. [35], who assessed the link between microelements and cognitive function, found that AD patients had lower plasma Se in comparison with control subjects. Other studies have shown that Se levels were positively correlated with cognitive function in elderly people, and thus Se deficiency could be a risk factor for AD [36,37,38]. During Se deficiency, the brain receives priority supplies, indicating the importance of Se in brain health. The turnover rate of some neurotransmitters in the brain has also been found to be affected by low Se status [39]. In addition, GPx enzymes are expressed in neurons and glial cells, where their free-radical-scavenging role protects against oxidative stress in the brain. GPx4 is synthesized endogenously and is the most widely expressed isoform in brain tissue. GPx4 has been recognized as a key regulatory factor in ferroptosis, a non-apoptotic and iron-dependent programmed cell death pathway, which causes a rapid elevation in ROS levels. Inhibition of ferroptosis by GPx4 provides protective mechanisms against neurodegeneration. Therefore, Se deficiency enhances susceptibility to ferroptotic processes, as well as other programmed cell death pathways, due to the decreased activity of GPx4 [40,41,42,43].

Studies have shown that Se status declines slightly in the elderly compared with younger adults [44,45,46]. The possible reasons include inferior bioavailability, increased requirements, metabolic changes, or a diet limited in energy, which may not be sufficient to provide adequate levels of microelements [45,47]. In our study, multiple linear regression analysis showed that frequent consumption of bread, butter, and, to a lesser extent, coffee, cheese, and tinned fish may be associated with increased concentrations of Se in the serum of patients with AD. The content of Se in food is closely related to its content in the environment. The concentration of Se in in Polish area, and thus in habitants is rather low [48]. However, food products from various regions of the European Union and other countries are available in stores. It should be remembered that, in excessive concentration, Se has a toxic effect. Se can oxidize endogenous sulfhydryl groups, increasing free radical formation [49]. In the present study, no significant correlations between Se and Zn concentrations and age were observed. Some studies have shown that during aging, the level of Zn in the serum decreases due to more limited food choices [50,51], which can cause inflammatory processes, increase oxidative stress, and decrease memory ability [52,53,54]. It has been discovered that proper Zn homeostasis in AD is important because an abnormally high concentration of this element induces the formation of granular tau aggregates in neuronal cells, tau aggregation-induced apoptosis and toxicity, contributing to the hyper-phosphorylation of tau. On the other hand, too low Zn levels lead to amyloid fibril development [55,56]. Our results regarding decreased serum Zn concentration are consistent with studies by other authors [30,32]. Xu et al. obtained different results: there were no differences in the concentration of Zn in AD patients compared to the control group, while the concentration of Zn in the plasma of men with AD was higher compared to healthy men [31]. In the present study, regression analysis showed that frequent consumption of flour products, honey, poultry, meat, jam, and tea may be positively correlated with serum Zn concentration, whereas frequent eating of cakes, sausages, raw vegetables, and cottage cheese may have a negative effect on the concentration of this trace element in the serum.

Oxidative stress plays an important role in AD etiology, and it can be the earliest event preceding this disease, although some structures formed in AD are related to the formation of ROS [57]. We observed significantly lower TAS in the serum of patients with AD compared to healthy people. However, both median values were within the reference range [27]. The sum of endogenous and food derived antioxidants represents the total antioxidant activity of the body. Our study shows that dietary habits may have a minor direct effect on serum TAS levels: only four dietary factors at 15% were included in the regression model. It has been shown that the cognitive functions of patients expressed by the MMSE scale positively correlate with TAS in the serum, and also significantly depend on the concentration of Cu in the serum. Patients with serum Cu levels above the reference values showed a significantly lower MMSE score, which is related to their worse clinical condition. We did not observe a difference in the serum Cu concentrations in AD patients compared to healthy controls, which is partially consistent with the results of other authors [31,32]. On the other hand, Vural et al. observed significantly lower Cu concentrations in people with AD, both in women and men [30]. Studies have shown that the expression of APP in vitro is dependent on the cellular Cu concentration at both gene and protein levels [58]. Formation of insoluble fibrils as a result of Aβ aggregation, the main pathological incident in AD, is a consequence of the interactions of Aβ metals, such as Zn and Cu [59]. Maintaining balance in all metal ions is important, but for Cu, it is crucial due to the fact that Cu is redox-active and can generate ROS involved in oxidative damage. It has been proven that Cu plays a critical role in the formation of β-sheet structures, which are considered to be an initial step of the formation of toxic aggregate of the Aβ fibrillar form. Therefore, Cu binding to Aβ has been recognized to play an important role in the neurotoxicity of Aβ [60,61]. In amyloid plaques and NFTs, a high concentration of Cu was found. NFT may be related to high concentrations of redox-active Cu [62]. Moreover, Cu through activating the cyclin-dependent kinase (CD K)5/p25 complex takes part in tau hyperphosphorylation. Some research indicates that GSK-3β kinase is also activated by Cu [63,64].

In our study, the Cu:Zn molar ratio was significantly higher in AD patients. An elevated Cu:Zn ratio is considered a marker of oxidative stress [65]. Relatively low levels of Zn and increased levels of Cu may impair the antioxidant properties of multiple enzymes [66]. Zn depletion and/or excess Cu levels enable NLRP3 inflammasome activation [67], and, consequently, the production of IL-1β in macrophages, which can lead to the initiation of an inflammatory process in AD [68].

Superoxide dismutase (SOD), a Zn- and Cu-dependent enzyme, is accountable for destroying free superoxide radicals in the body, and is a major participant in apoptotic signaling and oxidative stress. Decreased levels of Zn and Cu in the brain lead to abnormal functioning of SOD. Studies on transgenic mice have shown that in situations of cerebral oxidative stress, SOD is overexpressed, which might have a protective activity or an adverse effect on the central nervous system [69,70].

Our research has shown that fifteen nutritional factors can, to the extent of 38%, determine the concentration of Cu in the serum. Frequent consumption of cottage cheese, legumes and sausages may have a significant positive influence on serum Cu concentration, while the consumption of meat, honey, grits, rice, margarine, and coffee may significantly contribute to the reduction in Cu concentration in the serum.

Selected nutritional factors may affect the concentration of minerals in the body directly, being the source of the ingredient, or indirectly, by positively or negatively affecting its bioavailability from the diet [71]. Our research indicates that certain dietary modifications, changing the frequency of consumption of some food products, may improve the status of the examined antioxidant elements in people with AD. For example, regression analysis has shown that reducing the frequency of consumption of legumes, cottage cheese, and sausages could be beneficial. On the other hand, frequent consumption of flour products, honey, meat, and poultry may improve the mutual proportions between the examined elements, and thus have a beneficial effect on the antioxidant status.

Legumes have a high nutritional value and are recommended in reasonable nutrition. They are a source of proteins and fibers, and can contain variable amounts of mineral components. However, a limitation to legume consumption is related to their soaking and cooking processes. They can also be consumed in the form of processed products as canned legumes. Studies by Margier et al. showed that Zn content in legumes is 2.7–4.1-fold higher in relation to Cu, depending on the type and method of preparation [72]. Similarly, the content of Zn in cottage cheeses is approximately 3.7-fold higher than that of Cu [73]. Compared to the content of these micronutrients in cereals, the Zn to Cu ratio is more favorable—Zn content is 5.2–6.4-fold higher compared to the content of Cu [74]. Yu et al. indicated that whole wheat flour and wholegrain bread have better antioxidant properties than refined flour and white bread [75]. Likewise, Zn content in bee honey is much higher compared to that of Cu, for example, in buckwheat honey—by about 14 times [76]. In the Polish diet, meat accounts for 26% of the dietary intake of Zn [77]. However, the consumption of sausages as processed products containing preservatives should be limited due to possible carcinogenic effects [78]. Our research also shows an unfavorable Cu:Zn status in the case of frequent consumption of sausages.

The present study has some limitations. Firstly, the MMSE test was not performed in the control group. We also did not have data on BMI and smoking status in healthy subjects.

In summary, impaired homeostasis of essential minerals has significant effects, primarily on synapse function, induction of oxidative stress and inflammatory processes. Therefore, the maintenance of homeostasis and appropriate proportions of the minerals by modifying the diet of patients with AD seems significant.

## 5. Conclusions

In patients with Alzheimer’s disease, there is an abnormal proportion between antioxidant elements and decreased total antioxidant status, which may be associated with impaired cognitive functions. Therefore, it is important to provide adequate amounts of these antioxidants in the diet, especially products with a favorable ratio of Zn to Cu.

## Figures and Tables

**Figure 1 nutrients-13-00287-f001:**
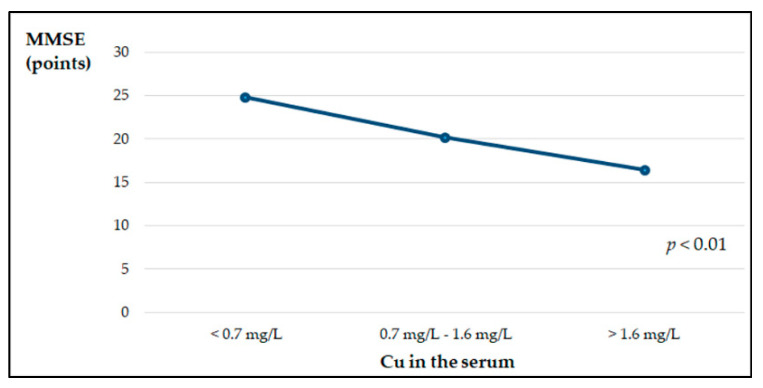
MMSE point scale depending on copper concentration in the serum of patients with AD, estimated by the ANOVA Kruskal–Wallis test.

**Figure 2 nutrients-13-00287-f002:**
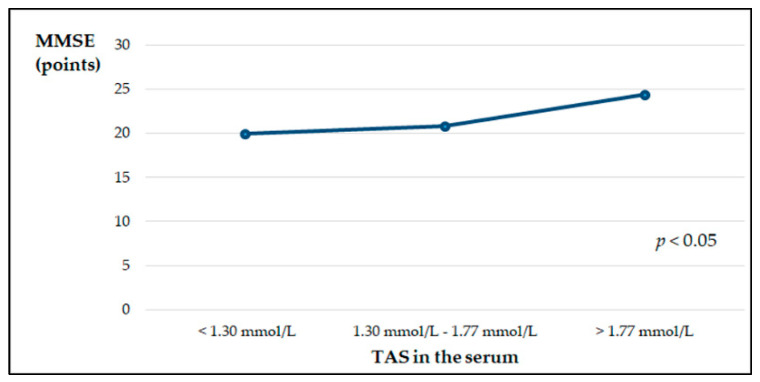
MMSE point scale depending on TAS in the serum of patients with AD, estimated by the ANOVA Kruskal–Wallis test.

**Table 1 nutrients-13-00287-t001:** Characteristic of the examined groups.

Variable	Control Group	AD Patients
Total (n)	60	110
Gender (M/F)	(14/46)	(30/80)
Age (years): average ± SD	67.0 ± 7.9	78.0 ± 8.1
Age (years): range	52–83	54–93
BMI (kg/m^2^): average ± SD	nd	26.5 ± 4.2
BMI (kg/m^2^): range	nd	17.8–40.2
MMSE (points): average ± SD	nd	20.4 ± 4.3
MMSE (points): range	nd	11–26
Smoking cigarettes * (n)/no-smoking (n)	nd	13/97
Alcohol drinking # (n)/no-drinking (n)	nd	5/105

AD—Alzheimer’s disease, n—number of subjects, M—male, F—female, SD—standard deviation, BMI—body mass index, MMSE—mini-mental state examination, nd—no data, * 5–10 cigarettes/daily, # more than once a week.

**Table 2 nutrients-13-00287-t002:** The concentration of selenium, copper, zinc, copper:zinc molar ratio, and total antioxidant status in the serum of patients with Alzheimer’s disease.

Variable	Control Group		AD Patients		*p*-Value
	(n = 60)		(n = 110)		
	Average ± SD				
	(min–max)				
	Median (Q1; Q3)				
	Total		Total		
	M (a)	F (b)	M (c)	F (d)	
Se (μg/L)	79.8 ± 22.0		69.1 ± 19.3		<0.05
	(44.4–138.4)		(18.27–117.62)		
	73.3 (65.6; 91.6)		71.4 (58.1; 79.7)		
	71.0 ± 13.9	81.7 ± 23.1	69.5 ± 17.9	68.9 ± 19.8	c vs. d *
	(52.4–92.9)	(44.4–138.4)	(41.7–107.5)	(18.3–117.6)	
	72.6 (58.5; 80.2)	73.3 (66.6; 100.2)	70.6 (52.1; 81.2)	72.3 (58.4; 79.4)	
Cu (mg/L)	1.07 ± 0.3		1.03 ± 0.23		ns
	(0.68–1.94)		(0.64–2.17)		
	1.02 (0.86; 1.24)		1.01 (0.87; 1.13)		
	0.96 ± 0.15	1.10 ± 0.30	0.90 ± 0.18	1.07 ± 0.23	c vs. d *
	(0.70–1.12)	(0.68–1.94)	(0.65–1.27)	(0.64–2.17)	
	0.97 (0.86; 1.09)	1.02 (0.85; 1.28)	0.87 (0.76; 1.05)	1.04 (0.96; 1.16)	
Zn (mg/L)	0.92 ± 0.23		0.75 ± 0.33		<00000.1
	(0.56–1.53)		(0.38–3.77)		
	0.86 (0.75; 1.02)		0.71 (0.62; 0.83)		
	0.88 ± 0.27	0.92 ± 0.21	0.84 ± 0.61	0.71 ± 0.14	c vs. d *
	(0.56–1.47)	(0.62–1.53)	(0.38–3.77)	(0.48–1.12)	
	0.84 (0.73; 0.93)	0.88 (0.75; 1.03)	0.75 (0.60; 0.90)	0.70 (0.63; 0.81)	
Cu:Zn molar ratio	1.24 ± 0.34		1.52 ± 0.47		<0.001
	(0.60–2.07)		(0.23–3.52)		
	1.20 (1.01; 1.48)		1.44 (1.26; 1.82)		
	1.21 ± 0.39	1.25 ± 0.34	1.31 ± 0.48	1.59 ± 0.45	b vs. d * c vs. d *
	(0.60–1.93)	(0.70–2.07)	(0.23–2.86)	(0.81–3.52)	
	1.23 (1.00; 1.34)	1.16 (1.01; 1.53)	1.33 (0.99; 1.50)	1.58 (1.33; 1.84)	
TAS (mmol/L)	1.43 ± 0.67		1.11 ± 0.42		<0.01
	(0.63–3.50)		(0.27–2.96)		
	1.31 (0.99; 1.70)		1.10 (0.83; 1.33)		
	1.88 ± 0.84	1.32 ± 0.59	1.17 ± 0.36	1.09 ± 0.44	a vs. b * a vs. c * b vs. d *
	(1.06–3.47)	(0.63–3.50)	(0.45–1.84)	(0.27–2.96)	
	1.69 (1.16; 2.41)	1.26 (0.94; 1.46)	1.23 (0.93; 1.43)	1.09 (0.79; 1.27)	

AD—Alzheimer’s disease, SD—standard deviation, Q1—lower quartile, Q3—upper quartile, *p*—level of significance, ns—no significant, M—male, F—female, a—male from control group, b—female from control group, c—male AD Patients, d—female AD Patients, TAS—total antioxidant status, * *p*-value < 0.05.

**Table 3 nutrients-13-00287-t003:** Stepwise multiple linear regression analysis of the influence of the frequency of consuming food products on the content of Se, Cu, Zn, and TAS in the serum of patients with AD: β coefficients and significance of variables entered in the model *.

Independent Variables	β Coefficient (SE)	Significance Level	Adjusted R^2^
Selenium			
White bread	0.371 (0.136)	0.0088	0.22
Wholegrain bread	0.292 (0.141)	0.0434	
Butter	0.295 (0.144)	0.0454	
Coffee	0.198 (0.125)	0.1210	
Cheese	0.148 (0.122)	0.2306	
Tinned fish	0.137 (0.117)	0.2485	
Legumes	−0.308 (0.125)	0.0169	
Tea	−0.226 (0.127)	0.0811	
Sausages	−0.175 (0.124)	0.1635	
Jam	−0.142 (0.124)	0.2581	
Copper			
Cottage cheese	0.360 (0.112)	0.0025	0.38
Legumes	0.347 (0.133)	0.0124	
Sausages	0.228 (0.113)	0.0501	
Wholegrain bread	0.259 (0.141)	0.0729	
Eggs	0.165 (0.116)	0.1624	
Butter	0.173 (0.131)	0.1925	
Fish	0.156 (0.119)	0.1961	
White bread	0.136 (0.125)	0.2811	
Meat	−0.490 (0.119)	0.0002	
Honey	−0.543 (0.152)	0.0009	
Grits, rice	−0.378 (0.125)	0.0040	
Margarine	−0.335 (0.133)	0.0152	
Coffee	−0.261 (0.113)	0.0254	
Vegetable oil	−0.205 (0.110)	0.0681	
Tinned fish	−0.143 (0.108)	0.1920	
Zinc			
Flour products	0.278 (0.129)	0.0358	0.26
Honey	0.266 (0.129)	0.0434	
Poultry	0.229 (0.123)	0.0692	
Meat	0.220 (0.123)	0.0791	
Jam	0.191 (0.129)	0.1433	
Tea	0.148 (0.119)	0.2170	
Cakes	−0.376 (0.117)	0.0023	
Sausages	−0.298 (0.130)	0.0264	
Raw vegetables	−0.183 (0.122)	0.1390	
Cottage cheese	−0.146 (0.120)	0.2297	
Total antioxidant status			
Flour products	0.378 (0.121)	0.0028	0.15
Ham	0.189 (0.128)	0.1453	
Poultry	0.119 (0.126)	0.3492	
Offal	−0.241 (0.118)	0.0457	

Se—selenium, Cu—copper, Zn—zinc, AD—Alzheimer’s disease, SE—standard error * Variables are listed in order of entry into the stepwise regression. Significance is set at *p* < 0.05.

## Data Availability

The data presented in this study are available on request from the corresponding author. The data are not publicly available due to privacy of ethical.

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
