# Peer review of "Dietary Habits, Selenium, Copper, Zinc and Total Antioxidant Status in Serum in Relation to Cognitive Functions of Patients with Alzheimer’s Disease"

_nutrients, 2021, doi:10.3390/nu13020287_

Round 1
Reviewer 1 Report
Dear Authors,
thanks for giving me the opportunity to read this Your manuscript.
The objective of Your study was to estimate selenium, copper, zinc and total antioxidant status in the serum of patients with AD in relation to their cognitive functions and dietary habits.
Here are my comments and suggestions:
- Introduction is too long and should be re-written, eliminating superfluous sentences (lines from 46 to 66) or repetitions (lines 68 and 69);
- ROS (line 74) is for ? As You know, acronyms must be specified when used for the first time ;
- Please write something about copper, as You did for selenium and zinc;
- Why was MMSE not "administrated" to the control group (see Table 1)? As You know, in clinical practice there are apparently healthy people having low MMSE scores;
- Table 2 and Figure 2 should be redone. Data regarding patients with early stage of disease were not presented;
- Discussion seemed weak. Indeed, Your discussion did not help to understand whether it was a casual association and/or a coincidence. For instance, You (and many of the Authors You mentioned) used a deductive methodology that has no scientific backing.....My personal opinion is that You could elucidate this question point better;
- You concluded that "increasing the supply of these antioxidants with the diet of patients is important". Well, as for today, this statement is little more than speculative.
Author Response
Dear Reviewer,
We would like to thank you for careful and thorough reading of our paper and for the thoughtful comments and constructive suggestions, which improve the quality of the manuscript.
We appreciate the time and effort that you dedicated to providing feedback on our manuscript.
We have incorporated your suggestions. Please see below for a point-by-point our response to your comments and suggestions:
Introduction is too long and should be rewritten, eliminating superfluous sentences (lines from 46 to 66) or repetitions (lines 68 and 69);
Thank you for this point. According to your suggestion, we have removed part of the introduction. However, it seems to us that some excerpts should remain, as we refer to them later in the discussion in terms of their relationship to the studied parameters.
ROS (line 74) is for ? As You know, acronyms must be specified when used for the first time ;
Of course, we agree with the comment. We are sorry for this mistake. We explained the ROS abbreviation on first use (line: 79-80).
Please write something about copper, as You did for selenium and zinc;
Thank you very much, we added some informations about copper. The role of copper has been described in connection with zinc, and also with iron in the context of its interaction with these elements in the brain. However, we agree with your remark that there was no separate information about copper (lines: 102-105).
Why was MMSE not "administrated" to the control group (see Table 1)? As You know, in clinical practice there are apparently healthy people having low MMSE scores;
The control group consisted of healthy people, most of them still professionally active (e.g. professors from our university), with no cognitive impairment, we added this information in the text (line: 145). However, you are right that conducting the MMSE test on healthy people would increase the value of the research. We added this fact in the limitation of the study (lines: 423-424). Currently we are unable to complete this data
Table 2 and Figure 2 should be redone. Data regarding patients with early stage of disease were not presented;
We observed a statistically significant difference between early and moderate AD stage only in the case of TAS. These results, together with the obtained values, have been included in the text in the part „Results” (lines: 242-244). In the case of other parameters, we did not found statistically significant differences between early and moderate AD (we have supplemented this information in the text -lines: 245-246). For this reason, we did not include this division of patients in the table and we included both groups of patients together.
Discussion seemed weak. Indeed, Your discussion did not help to understand whether it was a casual association and/or a coincidence. For instance, You (and many of the Authors You mentioned) used a deductive methodology that has no scientific backing.....My personal opinion is that You could elucidate this question point better;
In the discussion of the results, we tried to confront the obtained results with the results of other authors as much as possible and refer to the significance of the studied parameters in terms of brain function and their importance in AD. We are very sorry you felt that there was no scientific backing. For a better scientific explanation, we added certain mechanisms of action of the studied elements, including through their functions in antioxidant enzymes (lines: 315-325 and 387-392).
You concluded that "increasing the supply of these antioxidants with the diet of patients is important". Well, as for today, this statement is little more than speculative.
Thank you for this suggestion. Of course, we agree with your remark. We have changed this part of the conclusions to be more compatible with the obtained results concerning dietary habits.
The manuscript was additionally verified in terms of the English language by native speaker.
Once again, thank you for the time you put in reviewing our paper and for your suggestions.
Reviewer 2 Report
See my comments throughout the text of the attached PDF file.

Author Response
Dear Reviewer,
We would like to thank you for thorough reading of our paper, which improve the quality of the manuscript. We thank you very much for the positive comments. We appreciate the time and effort that you dedicated to providing feedback on our manuscript. We have incorporated your suggestions. Please see below for a point-by-point our response to your comments:
Some extra space should be added.
We added some extra space in the places you have marked.
Check grammar
We have corrected the sentence in the conclusions. The manuscript was additionally verified in terms of the English language by native speaker.
Once again, thank you for the time you put in reviewing our paper and for your recommendation for publication.
Round 2
Reviewer 1 Report
Dear Authors,
many thanks for your answers.
All my comments and suggestions were satisfatorily met.
My personal opinion is that significance of context, quality of presentation and scientific soundness have improved in the newer, revised version of your manuscript.